# Tissue Factor, Thrombosis, and Chronic Kidney Disease

**DOI:** 10.3390/biomedicines10112737

**Published:** 2022-10-28

**Authors:** Yuji Oe, Nobuyuki Takahashi

**Affiliations:** 1Division of Nephrology, Rheumatology, and Endocrinology, Tohoku University Graduate School of Medicine, Sendai 980-8574, Japan; 2Division of Nephrology and Hypertension, Department of Medicine, University of California San Diego, La Jolla, CA 92161, USA; 3VA San Diego Healthcare System, San Diego, CA 92161, USA; 4Division of Clinical Pharmacology and Therapeutics, Tohoku University Graduate School of Pharmaceutical Sciences & Faculty of Pharmaceutical Sciences, Sendai 980-8578, Japan

**Keywords:** uremic toxins, hypoxia, coagulation, protease-activated receptor

## Abstract

Coagulation abnormalities are common in chronic kidney disease (CKD). Tissue factor (TF, factor III) is a master regulator of the extrinsic coagulation system, activating downstream coagulation proteases, such as factor Xa and thrombin, and promoting fibrin formation. TF and coagulation proteases also activate protease-activated receptors (PARs) and are implicated in various organ injuries. Recent studies have shown the mechanisms by which thrombotic tendency is increased under CKD-specific conditions. Uremic toxins, such as indoxyl sulfate and kynurenine, are accumulated in CKD and activate TF and coagulation; in addition, the TF–coagulation protease–PAR pathway enhances inflammation and fibrosis, thereby exacerbating renal injury. Herein, we review the recent research studies to understand the role of TF in increasing the thrombotic risk and CKD progression.

## 1. Introduction

The number of patients with chronic kidney disease (CKD) has been increasing globally. The worldwide prevalence of CKD increased by approximately 29.3% between 1997 and 2017 [1]. CKD often results in renal replacement therapy and exacerbates the risk of stroke and cardiovascular events; therefore, patients with CKD have a poor prognosis and reduced quality of life [2,3,4].

Coagulation abnormalities are, generally, observed in patients with CKD and their risk of thrombosis is high [5]. The expression of markers associated with coagulation activities, such as von Willebrand factor, fibrinogen, factor VII (FVII), factor VIII, and D-dimer, is elevated in patients with CKD or those undergoing hemodialysis [6,7,8,9]. Patients undergoing dialysis have a higher mortality rate for pulmonary embolism, atherothrombosis, myocardial infarction, and stroke [10,11]. Reports have suggested that the risk of venous thromboembolism (VTE) increases by approximately 1.5–1.7 fold in patients with CKD and reduced renal function, such as those with stages 3 and 4 [12,13,14]. Another study showed an independent association of low estimated glomerular filtration rate (eGFR; <45 mL/min/1.73 m^2^) with VTE risk [15]. 

Tissue factor (TF, factor III) regulates the extrinsic coagulation system. TF dysregulation is linked to abnormal hemostasis and increases the risk of thrombotic events [16]. Typically, inflammatory mediators, such as cytokines, oxidative stress, and oxidized low-density lipoprotein, elevate TF levels, but recent studies have revealed the relationship between CKD-specific pathogenesis, including uremic toxins, and regulation of TF activity [17,18,19,20]. Therefore, new mechanisms underlying thrombosis development in patients with CKD are attracting increasing attention.

In addition to the pro-coagulant ability of TF, coagulation proteases, such as activated factor VII (FVIIa), factor X (FXa), and thrombin, which are located downstream of TF, activate protease-activated receptors (PARs) [21]. Signaling through PARs increases inflammation and kidney injury [21,22,23,24]; therefore, TF is a risk factor for thrombosis as well as an aggravating factor for renal injury. 

Herein, we review recent findings related to TF and coagulation in the pathogenesis of CKD to understand the role of TF in increasing thrombotic risk and CKD progression.

## 2. TF and the Extrinsic Coagulation System

TF is a transmembrane protein with a molecular weight of 47 kDa and an initiator of the extrinsic coagulation system. It binds to FVII to convert FVIIa. The TF/FVIIa complex catalyzes the activation of FX and factor XI (FXI). FXa and activated cofactor V (FVa) form a prothrombin complex, which generates thrombin. Finally, thrombin converts fibrinogen to fibrin to form thrombi [16].

To prevent abnormal activation of coagulation, TF is not normally expressed in cells exposed to blood, such as endothelial cells or monocytes, but is highly expressed in subendothelial cells, including vascular smooth muscle cells (VSMCs) [16,25]; however, under pathological conditions, such as inflammatory diseases and dyslipidemia, activator protein 1 and nuclear factor-κB (NF-κB) increase TF gene expression in monocytes and endothelial cells, and this activates the extrinsic pathway, resulting in a thrombotic tendency [16,26,27]. 

## 3. Expression of TF in the Kidneys 

In addition to the cells that make up blood cells and blood vessels, the expression of TF in kidney-derived cells has been previously characterized. In a study using human biopsy, the presence of immunoreactive TF was detected in podocytes and parietal epithelial cells [28,29]. Another study revealed that the level of TF in podocytes was increased via NF-κB signaling under hypoxic conditions [30]. In other kidney-derived cells, TF expression could be induced under pathological conditions; the expression of mesangial TF was increased in models of mesangial proliferative glomerulonephritis [31,32]. In addition, in vitro studies have shown that advanced glycation end products and elevated glucose levels upregulate TF expression in human mesangial cells [33]. Renal tubular cells also express TF, and the expression has been reported to be upregulated in models of unilateral ureteral obstruction and cisplatin-induced nephrotoxicity [34,35]. In vitro studies, various stimuli, including lipopolysaccharide, high glucose, and agonists of PARs, induce TF expression in renal tubular cells [36,37,38]. 

## 4. Relationship between Uremic Toxins and TF 

Uremic toxins are harmful metabolites that accumulate in the body of patients with CKD. More than 100 uremic toxins have been identified thus far [39], and the pathological roles of protein-bound uremic toxins, such as indoxyl sulfate, have been actively studied. Accumulated uremic toxins are involved in CKD progression and complications, including cardiac dysfunction, sarcopenia, and cognitive function [19,40,41]. Notably, the relationship between the hemostasis system and uremic toxins in CKD pathogenesis has recently received considerable attention [17,18,19,20] (Figure 1). 

### 4.1. Indoxyl Sulfate (IS) and Indole-3-Acetic Acid (IAA)

Indolic uremic solutes, such as IS and IAA, are tryptophan-derived uremic toxins. Indole is derived from the metabolism of tryptophan by intestinal bacterial tryptophanase and is absorbed through the intestine and metabolized to IS by cytochrome P450 2E1 and sulfotransferase in the liver. Similarly, IAA is metabolized from tryptophan by the gut microbiota [42,43,44]. 

IS and IAA have been reported to increase TF expression in endothelial cells, monocytes, and VSMCs [45,46,47,48]; moreover, indolic uremic solutes have been reported to induce the release of endothelial microparticles with TF pro-coagulant activity [46,49]. Mechanistically, indolic uremic solutes are ligands of the transcription factor aryl hydrocarbon receptor (AHR) [50]. Activated AHR translocates into the nucleus and regulates the expression of various genes, including that of TF. It has been reported that small interfering RNA or AHR inhibitors reduce the IS- and IAA-induced upregulation of TF expression in human endothelial cells and peripheral blood mononuclear cells [46]. Another study showed that IAA upregulated TF expression in human umbilical endothelial cells via the AHR/p38 mitogen-activated protein kinase (MAPK)/NF-κB pathway [51]. In addition, we demonstrated that IS and IAA induced TF expression in monocytic THP-1 cells, and this was suppressed by a MAPK inhibitor [47]. In vitro studies using VSMCs have revealed different mechanisms underlying the regulation of TF expression. Uremic serum, IS, and IAA upregulate TF levels by decreasing TF ubiquitination and increasing TF stability in VSMCs [52]. The IS–AHR pathway reduces the interaction between TF and the ubiquitin ligase STIP1 homology and U-box-containing protein 1 (STUB1), resulting in the inhibition of TF ubiquitination in VSMCs [48,53]. The IS-AHR axis may also induce TF expression by downregulating the Mas receptor expression or upregulating the pro-renin receptor expression in VSMCs [54,55]. The procoagulant effect of IS has also been demonstrated in animal studies; IS exposure increased arterial thrombotic tendency in rats [56,57] and mice [48], and this was suppressed by AHR inhibition.

### 4.2. Kynurenine (Kyn)

The Kyn pathway, another route of tryptophan metabolism, is catalyzed by the enzymes tryptophan dioxygenase (TDO) in the liver and indoleamine 2,3-dioxygenases (IDOs), which are inducible enzymes under pathological conditions, such as inflammation [58]. Kyn and its degradation products accumulate as a result of impaired renal function and play crucial roles in thrombotic activity during CKD pathogenesis [17,18,59]. In a previous study, it was demonstrated that the levels of plasma TF and other pro-coagulant markers were correlated with metabolites related to the Kyn pathway in patients undergoing dialysis or conservative treatment [60,61]. A study revealed that Kyn increased pro-coagulant TF activity in human VSMCs and arterial thrombosis formation in a carotid artery ferric chloride (FeCl_3_) injury model [62]. In addition, Kyn-induced TF expression and thrombotic activity are suppressed by AHR inhibition [62]. It has also been demonstrated that inhibition of IDO1 expression reduced blood Kyn levels, accompanied by a reduction in TF levels and thrombotic activity in a CKD model [63]. Moreover, the prothrombotic effect of Kyn on endothelial cells and venous thrombosis models via AHR has also been demonstrated [64]. Therefore, the Kyn–AHR–TF pathway is involved in the thrombotic tendency observed in patients with CKD.

### 4.3. p-Cresyl Sulfate (PCS)

PCS originates from phenolic compounds, such as p-cresol, which are converted from tyrosine and phenylalanine via bacterial fermentation [65]. Animal and human studies have demonstrated PCS to be associated with vascular dysfunction and cardiovascular events in CKD [66,67]; however, the effect of PCS on the induction of TF expression is still unclear. In vitro studies have shown that PCS fails to upregulate TF expression in human endothelial cells and monocytes [45,47]. In contrast, PCS increases the release of endothelial microparticles associated with thrombogenicity [68].

### 4.4. Trimethylamine N-Oxide (TMAO)

Trimethylamine (TMA) is generated by the gut microbiota from foods containing nutrients, such as choline, carnitine, and phosphatidylcholine. TMAO is biosynthesized from absorbed TMA by hepatic flavin-containing monooxygenases-3 in the liver [43]. Human studies have demonstrated that TMAO accumulates in the blood as renal function declines and that higher blood levels of TMAO are associated with CKD progression, CVD events, and mortality [69,70,71,72]. Prothrombotic roles of TMAO have been addressed in the literature; TMAO induces TF expression in human microvascular endothelial cells in a dose-dependent manner. An increase in aortic TF expression by TMAO was observed in choline-supplemented mice. In addition, TMAO enhances arterial thrombosis, which is inhibited by a TF-inhibitory antibody or a mechanism-based microbial choline TMA-lyase inhibitor in mouse models of arterial injury [72]. Another study showed that TMAO increased TF expression and activity via NF-κB signaling in vascular endothelial cells. In addition, TMAO and pro-inflammatory molecules, such as tumor necrosis factor α or high mobility group box 1 protein, synergistically increase TF expression in endothelial cells, suggesting its prothrombotic roles under inflammatory conditions [73].

## 5. Elevated Uremic Toxin-TF Axis and Risk of Thrombosis in CKD

A human study showed that blood TF levels were elevated with lower eGFR in patients with CKD and healthy controls [74]. In addition, blood TF levels were closely correlated with the levels of uremic toxins, such as IS and IAA, in patients with CKD [46,51]. Therefore, the uremic solute-TF axis, which increases with renal impairment, plays a crucial role in the thrombotic events in CKD. A previous study showed that a high serum IS level is associated with dialysis graft thrombosis after endovascular interventions [75]. Moreover, the Dialysis Access Consortium Clopidogrel Prevention of Early AV Fistula Thrombosis trial (DAC-Fistula) examined thrombotic complications after hemodialysis arteriovenous fistula creation. Participants with subsequent arteriovenous thrombosis had significantly higher levels of indoxyl sulfate and Kyn, and increased AHR and TF activity compared to those without thrombosis [62]. Similarly, subgroup analysis of the Thrombolysis in Myocardial Infarction II trial (TIMI-II trial), in which patients underwent percutaneous transluminal coronary angioplasty for myocardial infarction with ST-segment elevation, was performed. The thrombotic events in patients with milder CKD (stages 2 and 3) were associated with increased blood Kyn levels and AHR and TF activities [62].

## 6. Targeting Uremic Toxins for the Prevention of TF Activity and Thrombotic Risk

### 6.1. Role of Gut Microbiota in Reducing Uremic Toxins

Dysbiosis and alternation of the gut microbiota are observed in patients with CKD. For example, there is a decrease in the number of *Lactobacillus* groups, which are considered good bacteria, and an increase in the bacterial flora that contain urease, uricase, and tryptophanase (for example, that accelerates the production of toxins, such as indole or p-cresol) [76,77]. Dysbiosis has been suggested to be associated with the progression of renal damage as well as with cardiovascular complications [71,78,79]. Based on the relationship between TF and uremic toxins, intervention in the gut microbiota may help reduce uremic toxins and the risk of thrombosis.

Probiotics and prebiotics are typical therapies targeting intestinal bacterial function and composition. The former replenishes live microorganisms that have health benefits by improving and restoring intestinal microflora. The latter provides non-digestible compounds in foods that stimulate the growth and activity of beneficial bacteria, such as *Bifidobacteria* and *Lactobacilli* [80]. The impact of these approaches of the gut microbiota intervention on uremic toxins in patients with CKD has been demonstrated. Although the number of cases analyzed was small, probiotics reduced serum uremic toxin levels, such as IAA-O-glucuronide, in patients undergoing dialysis [81]. Another randomized study revealed that probiotic treatment for 24 weeks resulted in a reduction in plasma IS levels in patients undergoing dialysis [82]. These could be new therapies targeting uremic toxins; however, their effects on TF and coagulation activity, as well as on the risk of thrombosis, are unknown and require further investigation.

### 6.2. AST-120: An Oral Adsorbent of Uremic Toxin

AST-120 (Kremezin^®^) is an orally available adsorbent composed of porous carbon microspheres. These particles adsorb uremic toxin precursors in the intestinal lumen, allowing them to be expelled with stool [83]; thus, studies have shown that AST-120 ameliorates the accumulation of blood uremic toxins in patients with CKD [84,85] and in rat CKD models [86]. In addition, this drug reduces the accumulation of IS and PCS in several organs, including the kidney, skeletal muscle, and brain [87]. Whether the suppression of uremic toxins by AST-120 can reduce renal failure has also been examined; however, randomized controlled trials failed to demonstrate a treatment effect on the progression of renal impairment [88,89,90]. With regard to CVD complications, a basic study showed that AST-120 inhibited the progression of atherosclerosis in apolipoprotein E-deficient mice that underwent subtotal nephrectomy, suggesting an inhibitory effect on CVD events [91,92]. Whether the reduction in uremic toxins by AST-120 has the potential to reduce TF expression and coagulation is interesting and deserves further investigation. 

## 7. Hypoxia and TF in CKD

Hypoxia is involved in the pathogenesis of CKD. Cortical hypoxia has been detected using blood oxygenation level-dependent magnetic resonance imaging in patients with CKD [93] or a rodent model of diabetic kidney disease (DKD) [94]. Hypoxia is sensed by hypoxia-inducible factor (HIF), a master regulator of oxygen homeostasis, that regulates the expression of several genes to adapt to hypoxic conditions [95]. Under normal conditions, HIF-α, which is prolyl hydroxylated by the proline hydroxylase domain, is recognized by the von Hippel-Lindau E3 ubiquitin ligase complex and undergoes ubiquitination and proteasomal degradation; however, during hypoxic conditions, hydroxylation of HIF-α is inhibited, and it translocates to the nucleus, dimerizes with HIF-β, binds to hypoxia response elements, and induces transcription of hypoxia-related target genes [95]. The pathological roles of HIF in CKD progression have been actively studied, and its protective and harmful roles have been demonstrated [96]. 

Hypoxia enhances blood coagulation and thrombotic events [97]. HIF has been shown to regulate the expression of factors in the coagulation-fibrinolytic system as a transcription factor. Induction of TF expression by HIF1α has been reported in cancer-related thrombosis [98]. It has also been shown that HIF2α suppresses tissue factor pathway inhibitor (TFPI) and exacerbates thrombosis [99], and that HIF1α/HIF2α may upregulate plasminogen activator inhibitor-1 expression and suppress the fibrinolytic system [100]. These responses to hypoxia exacerbate thrombosis; in vivo level analyses have also shown that thrombosis is exacerbated in rodent models reared under hypoxic conditions [101,102]. 

In recent years, hypoxia-inducible factor prolyl hydroxylase (HIF-PH) inhibitors have become popular treatments for renal anemia. This drug improves renal anemia by stabilizing HIF and inducing downstream erythropoietin (EPO) expression. Its therapeutic effects are comparable to those of conventional EPO-stimulating agents [103,104]. One side effect of concern with HIF-PH inhibitors is the risk of thrombosis; a recent meta-analysis including 30 studies comprising 13,146 patients reported an increased risk of thrombosis by HIF-PH inhibitors compared to conventional therapy (risk ratio of 1.31, 95% CI 1.05 to 1.63) [105]. The causes of thrombosis are diverse and involve not only elevated coagulation factors but also platelet activation and excessive elevation in hemoglobin levels. As noted above, the activation of HIF induces TF expression, which may contribute to the risk of thrombosis associated with this treatment. There are still many unknowns regarding these factors, and further analysis is required.

## 8. Targeting the Coagulation System Reduces Kidney Injury in CKD

In addition to its prothrombotic effects, there is a great interest in whether TF is involved in kidney injury in CKD. Coagulation proteases downstream of TF, such as FXa and thrombin, activate PARs. PARs are members of the G protein-coupled receptor superfamily, and four PAR proteins (PAR1-4) have been identified. PAR is activated when its N-terminus is cleaved by proteases and the new N-terminus acts as a ligand to transmit signals. Each PAR is activated by a specific coagulation protease. The TF-FVIIa complex activates PAR2, ternary TF-FVIIa-FXa complex targets PAR1 and PAR2, and thrombin targets PAR1, PAR3, and PAR4 [21,106]. Signaling through PARs activates NF-κB or MAPK, which increases cytokine/chemokine or pro-fibrotic mediators, thereby exacerbating inflammatory and fibrotic diseases, including CKD [21,23]. Anti-inflammatory effects via suppression of PARs by factor Xa or thrombin inhibitors, such as rivaroxaban and dabigatran, have also been reported [21]. TF, coagulation proteases, and PARs could be involved in CKD progression, and their roles in CKD have been examined primarily in animal studies (Figure 2).

### 8.1. Relationship among TF, eNOS, and Inflammation in DKD

DKD is a major etiology of CKD and is associated with increased TF expression and hypercoagulability [23]. Animal studies have shown that renal TF expression is increased in type 1 diabetic models, such as the streptozotocin model [107,108]. In addition, our series of studies demonstrated a relationship between TF, endothelial nitric oxide synthase (eNOS), and DKD [23,109,110,111,112]. eNOS is an endothelium-derived NO with vasorelaxant and anti-inflammatory properties [113,114]. Genetic polymorphisms of eNOS are associated with the prognosis of DKD [115] and eNOS-deficient diabetic mice exhibit severe glomerulosclerosis and massive proteinuria as a model of human advanced diabetic nephropathy [116]. Using Akita diabetic mice with various eNOS expressions (*eNOS*^+/+^, *eNOS*^+/−^, and *eNOS*^−/−^), we demonstrated that diabetic kidney injury worsened in response to reduced eNOS levels. Renal TF expression and activity are correlated with decreased eNOS levels and severe kidney injury [109]. In another study, we demonstrated that renal TF expression was elevated when a high-fat diet and eNOS deficiency were combined [111]. These findings indicate a crucial role of eNOS and dyslipidemia in hypercoagulability under diabetic conditions. A causal link between TF and diabetic kidney injury has also been demonstrated; short-term administration of anti-TF neutralizing antibodies reduces inflammatory markers in the kidneys of diabetic mice lacking eNOS [111]. We also tested the roles of FXa and PAR2 downstream of TF in eNOS-deficient mice [110], and showed that the administration of a FXa inhibitor improved nephrosclerosis and inflammatory markers in diabetic Akita mice lacking eNOS. Similarly, PAR2 deficiency showed therapeutic effects in the same model. Another study showed that PAR1 and PAR2 inhibitors improved kidney injury in Akita mice with reduced eNOS expression and that the combination of these inhibitors additively reduced albuminuria and glomerulosclerosis with a reduction in inflammation and fibrosis markers [112]. 

### 8.2. Deletion of Myeloid TF Reduces Kidney Injury in Adenine-Induced CKD Model

CKD progression is characterized by the loss of renal cells and accumulation of the extracellular matrix. Adenine-induced nephropathy causes damage to the tubular epithelium, resulting in inflammation and interstitial fibrosis; therefore, it is widely used as a model of CKD [117]. Adenine-induced CKD is associated with an increase in TF expression in the kidney and a prothrombotic state [118]. We demonstrated the role of monocyte/macrophage TF in this model; conditional knockout of myeloid TF attenuated histological damage, including tubular atrophy, and the expression of pro-inflammatory cytokines in the kidneys of mice with adenine-induced renal fibrosis [47]. Further studies have also shown an inhibitory effect of factors downstream of TF on CKD, as FXa inhibitors have been shown to improve renal fibrosis in models of unilateral ureteral obstruction (UUO) and 5/6 subtotal nephrectomy [119,120]. Dabigatran, a direct thrombin inhibitor, ameliorated the epithelial–mesenchymal transition in the UUO model [121]. In addition, studies have shown that pharmacological inhibition or deletion of PAR1 and PAR2 protects against tubular injury in renal fibrosis models, including UUO, and adenine treatment [122,123,124,125,126]. Therefore, the TF-dependent coagulation and PARs pathways are involved in inflammation and pro-fibrotic response, which are hallmarks of CKD pathogenesis. 

### 8.3. Role of TF in Podocyte Injury

Podocytes and glomerular epithelial cells play crucial roles in the glomerular filtration barrier [127]. Podocyte injury is involved in the pathogenesis of many forms of glomerulopathy, including focal segmental glomerulosclerosis, DKD, and IgA nephropathy, and podocyte loss is an important factor in CKD progression [128,129]. As described above, TF is expressed in human podocytes. In addition, fibrin deposition in renal biopsy specimens is widely observed in glomerular disorders, suggesting a pathological role for intraglomerular coagulation in CKD [130]. An in vitro study using human immortalized podocytes demonstrated the harmful effects of TF in hypoxic podocyte injury. TF knockdown prevented the decrease in CD2-associated protein levels and actin reorganization under hypoxic conditions, suggesting that TF promotes reorganization of the actin cytoskeleton in podocytes and their injury [30]. In contrast, a unique role of TF in podocyte injury was shown; mice lacking the cytoplasmic domain of TF exhibited increased proteinuria under normal conditions or when nephritis was induced, accompanied by a reduction in the number of glomerular podocytes. An electron microscopy study demonstrated abnormal podocyte morphology due to a deficiency in cytoplasmic TF [131]. These findings suggest that the cytoplasmic domain of TF may contribute to the structural integrity of podocytes. 

## 9. Conclusions and Future Perspectives

This review outlines two possible effects of TF in patients with CKD, namely, the risk of thrombosis and the mechanism of renal damage, suggesting that TF is a promising target for the prognosis of CKD. Inhibition of abnormally activated TF remains a challenge, and reducing uremic toxin levels using probiotics or AST-120 can be used as a treatment option; however, this finding requires further validation in clinical studies. Recently, AHR inhibitors have been developed and are expected to be applied to cancer therapy [132]. They may help to prevent the risk of thrombosis caused by uremic toxins. Moreover, the cell-specific effects of TF in renal disease remain elusive. For example, conflicting effects of TF on podocyte damage have been reported. The mechanisms by which podocyte- and tubule-derived TF or the cytoplasmic domain of TF affect renal injury need to be determined to facilitate the use of TF as a therapeutic target for CKD.

## Figures and Tables

**Figure 1 biomedicines-10-02737-f001:**
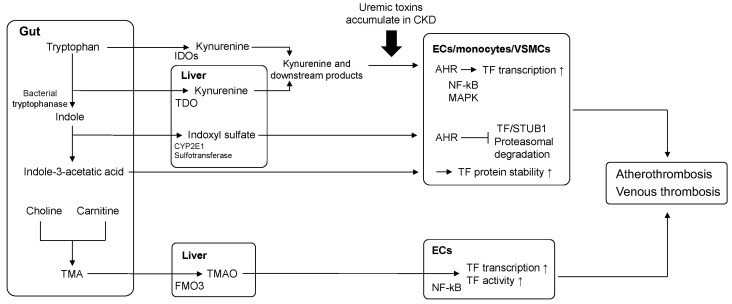
Uremic toxins increase TF in CKD. Kynurenine (Kyn) and related products are synthesized from dietary tryptophan by tryptophan dioxygenase (TDO) and indoleamine 2,3-dioxygenases (IDOs). Tryptophan is also converted to indole by bacterial tryptophanases and metabolized to indoxyl sulfate (IS) and indole-3-acetic acid (IAA). Tryptophan-derived uremic toxins accumulate in the body as renal function declines; they stimulate the aryl hydrocarbon receptor (AHR) and increase TF transcription. AHR also decreases the interaction of TF with the ubiquitin ligase STIP1 homology and U-box-containing protein 1 (STUB1), which inhibits TF ubiquitination and promotes TF stability. Foods containing nutrients, such as choline and carnitine, can be metabolized to trimethylamine (TMA) by the gut microbiota. Trimethylamine N-oxide (TMAO) is biosynthesized from absorbed TMA by hepatic flavin-containing monooxygenase-3 (FMO3). TMAO increases TF expression and activity in endothelial cells (ECs). VSMCs: vascular smooth muscle cells; and CYP2E1: cytochrome P450 2E1.

**Figure 2 biomedicines-10-02737-f002:**
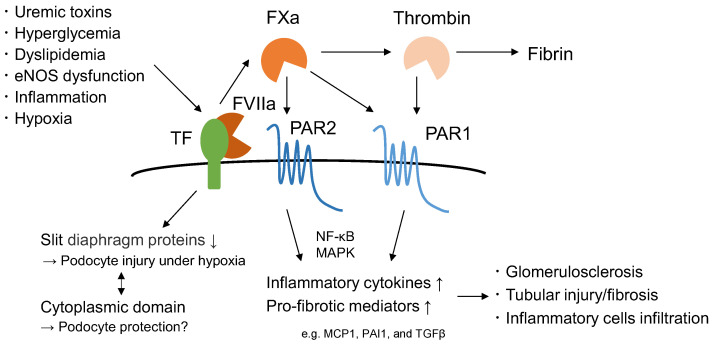
TF is involved in CKD progression. TF is activated by a variety of stimuli, including uremic toxins, hyperglycemia, and endothelial damage, such as eNOS dysfunction, under CKD pathogenesis. TF can activate downstream coagulation proteases, such as factor VIIa (FVIIa), factor Xa (FXa), and thrombin, which activate protease-activated receptors (PARs). PARs increase production of pro-inflammatory and pro-fibrotic mediators via NF-κB and MAPK signaling, resulting in kidney injury. TF reduces the expression of slit diaphragm proteins in podocytes and is, possibly, involved in podocyte injury under hypoxia; in contrast, the cytoplasmic domain of TF may contribute to the maintenance of normal podocyte structure and function. MCP1: Monocyte chemoattractant protein-1; PAI1: plasminogen activator inhibitor-1; and TGFβ; transforming growth factor β.

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
