# Peer review of "Tissue Factor, Thrombosis, and Chronic Kidney Disease"

_biomedicines, 2022, doi:10.3390/biomedicines10112737_

Round 1

Reviewer 1 Report

In general, this review article provides a very clear and comprehensive summary of the recent understanding of the role of tissue factor (TF) in thrombotic risk and chronic kidney disease (CKD). Particularly, this manuscript summarizes the mechanism of actions of a panel of uremic toxins, as well as the role of TF-PAR pathway in CDK progression. This is a well-constructed review, the literature is carefully described, and the different issues are presented in a well-organized manner. To move this manuscript forward to publishable level, I suggest to slightly modify Figure 2, to distinguish different molecules (such as FXa vs Thrombin; PAR1 vs PAR2) by using different colors/shapes.

Author Response

Thank you for your comments. We have colored them in Figure 2 to make it easier for readers to understand.

Reviewer 2 Report

The authors present a well-written and detailed comprehensive review of the role of tissue factor signaling in chronic kidney disease. The relationship between coagulation signaling and the mechanisms underlying are becoming increasingly recognized, and the pathogenic effects of tissue factor expression have been recognized in multiple disease states. Thus, this manuscript provides a comprehensive discussion of a timely topic that is likely to be of relevance and interest to a broad range of readers.

The suggested edits are minor.

Section 3
Line 77: Should define AGEs.

Section 4

Line 119: should define HUVECs

Line 170: Correct to HMGB1. Should define HMGB1 and TNFα.

Section 6

Any potential role for AhR antagonists? Would be worth mentioning.

Section 7

Line 245- The PH in HIF-PH is not specifically defined, although PHD is.

Section 8

Line 263-264- Need to reference this information about what activates the various PARs.

Suggest placing Figure 2 at the end of section #8, ie; after all the various components are explained/discussed in the body of the text.

Section 8.3:  Suggest amending the subsection title to something more applicable overall. For example “Targeting the coagulation system reduces kidney injury in CKD” or “Pharmacologic manipulation of the coagulation system in CKD”

Lines 313-314: Worth reiterating that Rivaroxaban targets inhibition of the Tissue Factor-Factor VIIa-Factor Xa complex.

Effects of fVIIa inhibitors?

Author Response

Thank you for your constructive comments.

Abbreviations have been corrected as you suggested.

1) Any potential role for AhR antagonists? Would be worth mentioning.

Response: AHR inhibitors are advancing in the field of oncology, and their application in the treatment of uremic thrombosis was described in section 9.

2) Line 263-264- Need to reference this information about what activates the various PARs.

Response: A reference was added.

3) Suggest placing Figure 2 at the end of section #8, ie; after all the various components are explained/discussed in the body of the text.

Response: We will ask the publisher to address your comment.

4) Section 8.3:  Suggest amending the subsection title to something more applicable overall. For example “Targeting the coagulation system reduces kidney injury in CKD” or “Pharmacologic manipulation of the coagulation system in CKD”

Response: We corrected the subsection title as “Targeting the coagulation system reduces kidney injury in CKD”.

5) Lines 313-314: Worth reiterating that Rivaroxaban targets inhibition of the Tissue Factor-Factor VIIa-Factor Xa complex. Effects of fVIIa inhibitors?

Response: We added a description in section 8 that TF-FVIIa-FXa complex targets PAR1 and 2, and  FXa inhibitors, such as rivaroxaban, inhibit them.  FVIIa inhibitors are also being developed, but their role in preventing kidney injury is not well demonstrated and was not included in this review.

Reviewer 3 Report

The manuscript is well written and describes several aspects of tissue factor in relation to thrombotic risk and chronic kidney disease.

Author Response

Thank you for reviewing our manuscript.